# Nanopore Long-Read Sequencing as a First-Tier Diagnostic Test to Detect Repeat Expansions in Neurological Disorders

**DOI:** 10.3390/ijms26072850

**Published:** 2025-03-21

**Authors:** Eddy N. de Boer, Arjen J. Scheper, Dennis Hendriksen, Bart Charbon, Gerben van der Vries, Annelies M. ten Berge, Petra M. Grootscholten, Henny H. Lemmink, Jan D. H. Jongbloed, Laura Bosscher, Nine V. A. M. Knoers, Morris A. Swertz, Birgit Sikkema-Raddatz, Dorieke J. Dijkstra, Lennart F. Johansson, Cleo C. van Diemen

**Affiliations:** Department of Genetics, University Medical Center Groningen, University of Groningen, 9713 CP Groningen, The Netherlands; e.n.de.boer@umcg.nl (E.N.d.B.); a.j.scheper@umcg.nl (A.J.S.); d.hendriksen@umcg.nl (D.H.); b.charbon@umcg.nl (B.C.); g.b.van.der.vries@umcg.nl (G.v.d.V.); a.m.ten.berge@umcg.nl (A.M.t.B.); p.m.grootscholten@umcg.nl (P.M.G.); h.h.lemmink@umcg.nl (H.H.L.); j.d.h.jongbloed@umcg.nl (J.D.H.J.); l.bosscher01@umcg.nl (L.B.); v.v.a.m.knoers@umcg.nl (N.V.A.M.K.); m.a.swertz@umcg.nl (M.A.S.); b.sikkema01@umcg.nl (B.S.-R.); d.j.dijkstra@umcg.nl (D.J.D.); l.johansson@umcg.nl (L.F.J.)

**Keywords:** short tandem repeat, long read sequencing, neurological disorders, spinocerebellar ataxia, fragile X syndrome, Oxford Nanopore Technologies, SNV, indel, methylation

## Abstract

Inherited neurological disorders, such as spinocerebellar ataxia (SCA) and fragile X (FraX), are frequently caused by short tandem repeat (STR) expansions. The detection and assessment of STRs is important for diagnostics and prognosis. We tested the abilities of nanopore long-read sequencing (LRS) using a custom panel including the nine most common SCA-related genes and FraX and created raw data to report workflow. Using known STR lengths for 23 loci in 12 patients, a pipeline was validated to detect and report STR lengths. In addition, we assessed the capability to detect SNVs, indels, and the methylation status in the same test. For the 23 loci, 22 were concordant with known STR lengths, while for the last, one of three replicates differed, indicating an artefact. All positive control STRs were detected as likely pathogenic, with no additional findings after a visual assessment of repeat motifs. Out of 226 SNV and Indel variants, two were false positive and one false negative (accuracy 98.7%). In all *FMR1* controls, a methylation status could be determined. In conclusion, LRS is suitable as a diagnostic workflow for STR analysis in neurological disorders and can be generalized to other diseases. The addition of SNV/Indel and methylation detection promises to allow for a one-test-fits-all workflow.

## 1. Introduction

While the inherited neurological disorders, spinocerebellar ataxia (SCA) and fragile X (FraX), can be caused by all types of genetic variants, they are most frequently caused by expansions of short tandem repeats (STRs) [1,2,3]. STRs are highly polymorphic 2-6 base pairs motifs consecutively repeated at a given genomic position and varying in length between unrelated individuals [4]. The detection of STR expansions and interruptions with alternative motifs is important in SCA and FraX for establishing disease age-of-onset, prognosis, and severity [3]. The current main method of choice—short-read sequencing (SRS)—can detect these expansions for most genetic referrals when combined with tools such as ExpansionHunter [5]. However, the sizes reported are inaccurate [6,7], so several different PCR techniques are currently used to accurately measure the expansion length and motif requiring always multiple PCR reactions per patient. Conducting these separate assays significantly increases workload, as each requires distinct implementation pathways to incorporate additional genes into diagnostic workflows. As a consequence, the number of genes in which STRs can be detected is limited. Yet new causal repeat expansions continue to be discovered in SCA, exposing the need for a generic, adaptable test for all the genes and motifs of repeat expansions [8].

One such technique is Oxford Nanopore Technologies long-read sequencing (LRS) (Oxford Nanopore Technologies (ONT), Oxford, UK). In contrast to SRS, ONT LRS’s longer reads (an average length of 10 kb) enables a detection of the full length and sequence of even highly expanded STRs [9,10]. An additional benefit of ONT LRS is that it can also simultaneously detect base modifications such as methylation in the sequenced DNA fragments [11,12]. These methylation patterns are relevant for predicting the pathogenicity of multiple repeat expansion diseases, because they indicate transcriptional repression [13]. For example, methylated *FMR1* repeats are a cause of FraX, whereas non-methylated *FMR1* repeats are not, although they can cause other conditions such as Fragile X-associated tremor/ataxia syndrome (FXTAS) [14,15]. A further benefit of using ONT LRS as a diagnostic test for neurological disorders is that it can also simultaneously detect single nucleotide variants (SNVs), indels, and structural variants (SVs) [16,17], removing the need for additional SRS analyses.

Our ultimate aim for genome diagnostics is a one-test-fits-all approach, that reduces the number of wet-lab methods needed. Here, we demonstrate the implementation of an ONT LRS diagnostic workflow for the detection of STR expansions in the neurological disorders SCA and FraX. We also measure ONT LRS’s detection accuracy for SNVs and indels and for *FMR1*-specific methylation to further assess its feasibility as a one-test-fits-all diagnostic approach.

## 2. Results

### 2.1. Validation Phase

In the validation phase, we compared the LRS results to the diagnostically reported repeat lengths. We produced 8.15–28.30 Gb per sample with a mean fragment length between 8 and 9 kb (Appendix A). With this yield, on average, 99.3% of the exons of the 10 SCA-related genes were covered at least 10× for 95% of the gene. For 20× and 30× coverage, these percentages were 77.2% and 31.4%, respectively. For the X-linked *FMR1* gene, 87.1% of the exons were covered at least 10× for 95% of the gene, and this was 33.8% and 12.9% at coverages of 20× and 30×, respectively (Appendix A). Coverages of the repeat regions differed between genes, ranging from 5 to 101 supporting reads (Table 1, Appendix A). All the samples tested were concordant between sequencing and OpenArray, excluding one sample swap (see Appendix A).

For all the repeat regions of the genes of interest, the repeat length was determined in all patient samples (Appendix A). For the 23 positive control repeat regions with a known STR length, 22 had no more than the predefined allowed difference in the number of RU between the diagnostic result and the LRS result (Table 2). In sample 5, LRS detected lengths of 26 and 30 RU while diagnostically reported STR lengths were 30 and 31 RU for *ATXN1* (Appendix A). However, in the replicate sample, LRS detected for both alleles a length of 30 RU, suggesting that the 26 RU call was a potential technical outlier. This was further supported by the fact that there were 2 and 31 reads supporting the call of, respectively, the 26 and 30 repeat units. This does not sufficiently support a heterozygous call with 26 repeat units when the coverage is 33 times. In the analysis pipeline these outlier calls can be avoided by increasing the thresholds for read support depending on the coverage. The other replicate samples gave identical results, indicating a high reproducibility.

### 2.2. Implementation Phase

In the implementation phase all samples were processed using the automated pipeline, without using prior knowledge, to design the procedure for use in diagnostics. For each sample, a report in a HTML format was generated in which variants could easily be assessed for use in a diagnostic setting. In the example VIP report shown in Figure 1, all known control expanded and non-expanded STRs were reported as VUS or LP, based on the repeat length (Table 3). Repeat expansions above the pathogenic threshold were detected in *FGF14* and *RFC1*, after which the repeat motif was checked visually in the CRAM file using IGV. Expansions with an alternative (non-pathogenic) motif were subsequently re-classified as LB. The number of reads supporting the calls are shown in Table 1. For some patients, there were no diagnostically reported control variants, but the pipeline did give automatic calls for VUS variants, mostly for *RFC1* (Table 3).

### 2.3. The Accuracy of ONT LRS for SNV- and Indel-Calling and Methylation Detection

#### 2.3.1. SNV and Indel Calling

To determine the accuracy of ONT LRS SNV- and indel-calling for the genes in our panel, we compared the ONT variant calls with the NGS variant calls for six samples. This identified 28 indels and 198 SNVs per sample that were concordant and 1 indel and 17 SNVs that were not (Table 4).

The non-concordant call could be explained by various characteristics: Twenty-five were located directly next to or inside a homopolymer or repeat region. In the sample that underwent QXT sample preparation (S1), 12 SNVs were not called in the WES data, because tagmentation resulted in reads starting at the same position, leading samtools markdup to label them as duplicate reads. For six variants, a discrepancy remained. Based on a manual inspection of the CRAM and vcf files, we deemed two a likely LRS false negative, one an LRS false positive, two a WES false negative, and one a WES false positive result (Appendix A). No confirmatory testing was performed for these variants.

#### 2.3.2. Methylation Detection

For the four samples with a FraX indication, we determined the methylation profile of *FMR1*. In the male control sample without a repeat expansion, only unmethylated reads were detected in the *FMR1* promoter region (Figure 2a). Due to X-inactivation in the female sample (S4) without a repeat expansion, half of the reads were methylated (Figure 2b). In the affected male (S6) with a known mosaic repeat expansion, some reads were methylated and some unmethylated (Figure 2c). In the female carrier (S9), all reads with a repeat expansion and some reads without an expansion were methylated (Figure 2d). We observed no clear relation between the repeat length and methylation status per sequencing read.

## 3. Discussion

Our main aim in this study was the implementation of ONT LRS in our diagnostic workflow for the detection of STRs in neurological disorders. In current diagnostics, multiple tests are needed, which makes it laborious. For our panel of genes, we show that STR lengths can be detected with a high accuracy using ONT LRS AS and that automatic classification using the VIP pipeline allows for an easy interpretation in a diagnostic setting. For 22 out of the 23 control STRs, the results were concordant between LRS and current diagnostic methods. Moreover, the discrepancy for the remaining STR was a likely technical artefact and did not alter the diagnostic outcome. Our data also suggest that even with limited coverage, accurate SNV- and indel-calling is within reach. All the discrepancies we observed between variants detected with LRS and SRS could be explained based on the variant location (e.g., next to a repeat) and/or variant quality score. On two occasions, we missed a variant in LRS due to an insufficient number of high-quality reads covering the region of interest. This issue would likely be solved by increasing the coverage to 30×, as already shown by others [17]. Finally, we show that the methylation status of the *FMR1* repeat region can be determined with LRS. This allows for an assessment of the level of methylation in case of the somatic instability of pre-mutations. Our results suggest that it is possible to implement a one-test-fits-all neurological disease panel that includes STR, SNV, indel, and methylation calling using LRS with an average coverage of 30×. Depending on the already available resources of laboratories, this will be or in the near future become cost-effective because more variant types will be detected with one single technique. A larger and more diverse cohort would be necessary to confirm the reproducibility and robustness of the findings across different laboratories and populations. Our results and conclusions are in line with earlier research performed on the detection of STRs, SNVs, indels, and methylation using ONT LRS adaptive sampling in highly similar patient groups [9,10]. Compared to these previous studies, we used automated STR-calling and classification which brings LRS-based STR analysis closer to clinical use.

An important observation arising from this study is that a thorough interpretation of the sequencing details was of the utmost importance to correctly check for concordance between the results of LRS and currently used diagnostic methods. For example, using a previous version of the Straglr algorithm, we initially observed a discrepancy between LRS results and the diagnostic result for *ATXN3*. The cause of this discrepancy is a difference in the interpretation of the actual nucleotide sequence of the repeat expansion. The *ATXN3* reference repeat consists of 11 GCT RU, interrupted by a GTT after 7 RU and a TTT after 9 RU. Our current diagnostic method (fluorescent fragment analysis) calculates the repeat length by subtracting a predefined fixed number of bases outside the total repeat (based on the known positions of the primer sequences used in the method), independent of the presence or absence of putative interruptions, from the measured fragment size based on Sequeiros et al. [18]. A previous version of the Straglr algorithm, however, did not include RU after an interruption in the repeat sequence was detected. We modified the Straglr catalogue to include the RU after the interruption, resulting in RU lengths determinations concordant to diagnostic results. During our studies, we encountered several such instances, indicating that LRS interpretation heavily relies on the used catalogue and manual curation of the catalogue.

The PCR-free LRS-based STR length determination has technical benefits compared to conventional PCR-based STR detection methods. The latter sometimes produces inaccurate repeat-length estimates: actual repeat lengths can be overestimated due to PCR stutter or underestimated because smaller fragments are easier to multiply [19,20]. The method is often seen as the gold standard against which a new method should be validated and tested for concordance, yet the new method may actually be more accurate. Because the estimations of the pathogenicity of repeats are based on current methods, these thresholds may need to be adapted for the more accurate STR-calling in the new method, particularly if the STR lengths consistently differ between both methods. Another advantage of LRS is the availability of the complete repeat sequence, including interruptions and pre- and post-repeat sequences. This can be highly important for STR diagnostics, as exemplified by the relevance of specific sequence motifs around the repeat as well as repeat interruptions for disease prognosis in *FGF14*-involved SCA [21].

LRS provides clinicians and laboratory specialists with previously un-accessible high resolution data. This leads to interpretation issues. One issue is the presence of somatic variation in repeat length. For instance, for *FMR1*, pre- and full mutations can be unstable and can thus differ in the number of RU between cells [22]. As the presence of longer or shorter repeats in part of the cells can affect diagnosis and prognosis, it is important to assess putative, somatic differences in STR length. Our VIP pipeline uses the median repeat length for automatic classification, but it also reports the range in the length of repeats of the individual reads. In one sample (S6), with a known mosaic *FMR1* repeat expansion (range 213−300 RU), VIP detected a 223 CGG repeat in *FMR1*. This was above the reported minimal pathologic threshold of 200, resulting in an automatic LP prediction. Our workflow was able to detect the mosaic because the range in RU was 147−397. This means that some of the STR sequences were >200 RU, indicating that a pre-mutation rather than a full mutation is present in some cells. The methylation status of the *FMR1* promoter also showed that only some of the reads were methylated, although some with a longer repeat remained unmethylated. As it has been shown that methylation mosaicism can be present alongside size mosaicism and that the *FMR1* promoter on chromosomes with a full repeat expansion can remain unmethylated [14] it is likely that such a mosaicism is present in this case.

An additional benefit of using ONT AS to detect STRs is its flexibility. New disease-associated genes or genomic regions can easily be added without altering the sample preparation with an updated enrichment kit. For example, SNV and indel variants in *FGF14* are associated with SCA27 [23], and a repeat expansion in FGF14 was recently shown to cause a SCA subtype, called SCA27b [24,25]. Adding STR detection for this gene to our panel was straightforward, whereas an extended lab validation was required to implement the PCR-based test. It has also been shown that the need for gene-specific assays has led to an underdiagnosis of repeat expansion disorders [26], with some STRs not being tested due to a too-limited yield per STR. Adding these rarer SCA-causing genes to the LRS panel would be instantly cost-effective as no additional lab tests or analyses are necessary. To increase diagnostic yield, we therefore suggest that a broad neurologic disorder gene panel including the rarer SCA-causing genes needs to be developed.

The automatic workflow we here describe in our paper produces a user-friendly standalone HTML report listing the results for all STRs of interest. This report allows users to show or hide repeats based on various parameters, such as the automatic classification (e.g., LP), thereby aiding rapid variant analysis. Furthermore, other variant types that could explain the patient’s phenotype can be included in the report in a separate or combined view. Because of the variable nature of the sequences, with pre- and post-repeats and interruptions with RU that have different nucleotide components, an automatic detection and classification of these motifs is difficult. Straglr and Stranger currently do not support an analysis of alternative RU, such as the AAG and AAGAGG RU in *FGF14*. Therefore, for expanded alleles, a visual assessment of the motifs is currently necessary for genes with known alternative motifs (e.g., *FGF14* and *RFC1*). However, future developments will allow for automatic motif detection, abolishing the need for manual re-analysis, thereby allowing for a wider adoption of the technique.

The only types of genetic variations not tested in this study are larger copy number variation and SVs other than STRs. However, ONT sequencing has been shown to have a relatively high sensitivity for such variants, from 50 bp up to several Mb [16,17]. Finally, we previously demonstrated that in the Netherlands, copy number variations (>50 kb) are only a very rare cause of SCA [27]. Therefore, we did not include the detection of these types of variants in this test. However, these types of variants will be important for other diseases and the detection and annotation of these variants can easily be added to our modular pipeline [28].

Overall, we have demonstrated that an LRS-based diagnostic workflow to detect causal STR expansions in neurological disorders is feasible. Here, we used it to replace SCA and *FMR1* diagnostics, but this method can be generalized to other diseases. We further believe that with increasing coverage, this workflow can be implemented in a one-test-fits-all workflow for the detection of SNVs/ indels, methylation patterns, and STR-repeats.

## 4. Materials and Methods

### 4.1. Sample Selection and Study Setup

We set up our neuro-STR-test to allow for the detection of STR expansions in neurological disorders for 10 genes for which STR analysis is commonly performed, nine causing SCA (*ATXN1*, *ATXN2*, *ATXN3*, *ATXN7*, *ATXN8*, *ATXN10*, *CACNA1A*, *RFC1*, and *FGF14*) and one causing FraX and FXTAS *(FMR1)*. These genes were selected based on Genereviews [8] and on their relevance to adult-onset SCA diagnostics and FraX.

To test the performance of the neuro-STR-test for a broad range of STR lengths, we defined three length categories: 1−50 repeat units (RU), 50−300 RU, and >300 RU. For each category, we selected samples from at least three patients carrying known, but differing expanded and non-expanded STRs, as previously determined using standard of care PCR-based diagnostic testing. To test the LRS method’s reproducibility, one sample per length category was replicated for the whole procedure.

In total, we selected 12 pseudonymized patient samples with 23 non-expanded and expanded STRs in *ATXN1* (4), *ATXN2* (3), *ATXN3* (4), *CACNA1A* (4), *ATXN7* (3)*, FGF14* (1), and *FMR1* (4). In our current diagnostics, SCA-associated repeats are determined with PCR fragment analysis (Appendix A) and FMR1 repeats are measured with the Amplidex PCR CE FMR1 kit (Asuragen, Austin, TX, USA).

For six patient samples, whole exome sequencing (WES) data was available. For five of these, data were produced by Illumina NovaSeq 6000 (Illumina, San Diego, CA, USA) with enrichment using the Exome v7 enrichment kit with SureSelect XT sample preparation (Agilent, Santa Clara, CA, USA). One sample was sequenced by Illumina NextSeq500 and enriched using the Exome V6 Enrichment kit with QXT sample preparation (Agilent).

All patients gave informed consent for the use of diagnostically obtained materials for innovations of diagnostic care and the study protocol was approved by the medical ethics review board of the UMCG (METc 2023/507).

### 4.2. Design of the Gene Panel for Adaptive Sampling

ONT allows for targeted LRS by ejecting off-target DNA-sequences, a method called adaptive sampling (AS). The optimal size of an AS panel is 1–5% of the genome. We included the neuro-STR-test in a broader panel of 468 genes from the neuro- and epilepsy-panels currently used in our SRS diagnostics. We used SureDesign (Agilent) to select the entire transcribed region of the genes of interest in Genome build UCSC hg38, GRCh38, December 2013 (National Center of Biotechnology information (NCBI), Bethesda, MD, USA) in combination with the databases RefSeq [29], Ensembl [30], and CCDS [31]. All SureDesign-selected regions were manually checked in these databases. The AS BED file contained the genomic regions of the (predicted) very first start and the very last stop of different isoforms of all genes of interest, all extended by 10 kb at both sides of these start and stop positions to increase the enrichment efficiency (Appendix A). An additional 57 SNPs (Appendix A), also flanked by 10 kb at both sides, were included in the AS BED file to allow for a concordance check. The total size was around 1.5% of the genome.

### 4.3. DNA-Extraction and Quality Control

Maxwell RSC (AS1400) (Promega, Madison, WI, USA) chemistry was used for DNA-extraction on a Hamilton NIMBUS Presto system (Hamilton, Reno, NV, USA and ThermoFisher, Waltham, MA, USA). The DNA length, quantity, and purity were checked with Nanodrop spectrophotometry (OD 260/280 ~ 1.8, OD 260/230 2.0–2.2) (ThermoFisher) and Qubit (DNA dilution 1:8) (ThermoFisher).

### 4.4. LRS Sample Preparation

The ONT protocol was started with 8 µg DNA. Short fragments were eliminated with the SRE XS kit (Pacific Biosciences (PacBio), Menlo Park, CA, USA) and the recovered DNA was eluted in 50 µl elution buffer. In contrast to the PacBio protocol, we washed 2 times with 70% ethanol. Fragments ≤ 10 kb were progressively depleted. Fragments ≤ 4 kb were almost completely removed. After the DNA measurement with Qubit (1:5 DNA dilution) (ThermoFisher), DNA was sheared with Covaris g-tube (2300 RCF, 1 min) (Covaris, Woburn, MA, USA), vaporized with speedvac (ThermoFisher) to 50 µL, and again measured with Qubit (1:5 DNA dilution) (ThermoFisher). Next, the SQK-LSK114 sample prep (ONT) was performed with an input of 4 µg sheared DNA, following ONT instructions. Incubation times and purifications steps with AMPure XP beads (included in SQK-LSK114) were amended to the following: 20 °C, 10 min and 65 °C, 10 min (DNA-repair and end-repair, Heated lid 75 °C); 5 min RT (60 µL AMPure XP beads); 10 min, 37 °C, 300 rpm (elution DNA-repair and end-repair); 20 °C, 20 min (adapter-ligation, heated lid off); 5 min RT (80 µL AMPure XP beads, long-fragment buffer); and 30 min, 37 °C, 300rpm (elution adapter-ligation). After the measurement of the DNA-concentration (1:3 DNA dilution) with Qubit (ThermoFisher) and of the DNA distribution with Genomic DNA ScreenTape (Agilent), NEB Bio Calculator (New England Biolabs, Ipswich, MA, USA) was used to measure the library concentration.

### 4.5. ONT Sequencing

An R10.4.1 flow cell (FLO-MIN114) with at least 1000 active pores was primed, loaded (70 fmol library), and run on a GridION device, following ONT instructions. The following settings were used as follows: pore scan frequency 1.5 h, reserved pores on, minimum read-length 200 bp, read-splitting on, high accuracy base calling, 400 bps. Dorado (ONT) fast base calling was used to select the regions of interest in the AS gene panel. Reads were aligned to the GRCh38_no_alt_analysis_set_GCA_000001405.15 reference genome (GRCh38) (NCBI).

### 4.6. Concordance Check

To rule out possible sample swaps, we performed a separate test on 11 samples using Open OpenArray (Thermo Fisher Scientific) to determine the genotypes for the 57 common SNPs added to the AS panel. A sample was considered concordant when at least 17 SNPs were informative and there was a 95% concordance between LRS and OpenArray.

### 4.7. Data Processing

The MinKnow software 23.11.7 (MinKnow core 5.8.6) (ONT) stored the squiggle (electric trace) in POD5 files (ONT). The squiggle was then converted to base calls. High accuracy base calling was done during sequencing using Dorado 7.2.13 (ONT) and reads passing quality metrics (Q-score > 9) were stored in FASTQ files. For each run (flow cell), an adaptive_sampling_*.csv file was produced. This file records which reads were rejected (unblock), accepted (stop receiving), or passed the pore before a decision was made (no decision). FASTQ files were merged and decompressed and subsequently read IDs of interest (stop receiving) were extracted and saved in a new FASTQ file.

We performed data analysis with the open-source MOLGENIS Variant Interpretation Pipeline (VIP) 7.9.0 [28]. In short, reads were aligned to a GRCh38 reference genome (NCBI) using minimap2 [32,33]. STRs were called using Straglr [34] and annotated with Stranger (https://github.com/Clinical-Genomics/stranger (accessed on 19 March 2025)) using a Straglr catalogue defining the region in the reference genome where the repeat is located, the expected repeat unit, and the related gene (Appendix A). In addition, a Stranger catalogue defining information on the inheritance mode and repeat length pathogenicity thresholds was created for each expected repeat unit (Appendix A). Straglr and Stranger were incorporated into VIP, which provided annotation and an automatic classification based on the maximum normal repeat length (NormalMax) and the minimum pathogenic repeat length (PathologicMin) defined in the Stranger catalogue. STRs with a length between those values were classified as a variant of unknown significance (VUS). STRs with a length above the pathogenic threshold were labelled likely pathogenic (LP). VIP output is shown in a user-friendly, portable html report (Figure 1), with variants shown based upon the selected decision tree (Appendix A, decision tree VIP). We then visually assessed the repeat patterns based on the reads in the CRAM file using IGV 2.16.2. We then visually assessed the repeat patterns based on the reads in the CRAM file using IGV 2.16.2 [35].

### 4.8. STR Expansion Detection: Validation

Our study of the neuro-STR-test encompassed several parts. In the initial validation phase, we compared the LRS results to the diagnostically reported repeat lengths.

This was followed by an implementation phase in which all samples were processed using the automated pipeline, without using prior knowledge, to design the procedure for use in diagnostics. SNV and indels were then compared with the Illumina SRS results to investigate whether it is possible to include calling these variants in the same test. Finally, we assessed the methylation profiles for patients with diagnostically reported *FMR1* repeats.

All validation samples were processed using VIP. If a sample was sequenced on two flow cells, it was processed both with all data combined and per flow cell.

In the validation phase, we compared the diagnostically determined STR lengths with the repeat lengths detected by the LRS workflow and measured the concordance between them using the difference in RU. We defined results as concordant when the maximum difference was ±2 RU for STRs of 1−50 RU, and ±3 RU for STRs of 50−300 RU, which correspond to the accepted variability between repeated tests in current diagnostics. As STR expansions >300 RU are classified as “>300” in current diagnostics, LRS results that also had an STR length >300 were considered concordant. For all discordant results, we re-assessed the diagnostic data to understand the reasons for the differences and determine which method provided the most reliable results. If both methods gave the same diagnostic result, we considered LRS to be a valid diagnostic technique.

### 4.9. STR Expansion Detection: Implementation

In the implementation phase, we used the output of our automated VIP STR workflow. Based upon the automatic classification, we collected all variants with at least one allele automatically classified as VUS or LP. For repeats known to have various RU motifs or interruptions (*FGF14, RFC1,* and *ATXN1*), we manually assessed the expanded repeats, to determine if the pathogenic or benign motif was present. The presence of a benign RU led to a manual reclassification to likely benign (LB). For *FMR1* samples, we executed an extra pipeline (see methylation calling) to detect 5mC base modifications, with a visual assessment using IGV.

### 4.10. SNV and Indel Detection

We also tested the detection accuracy of ONT LRS AS for SNVs and indels to assess its feasibility as a single method for SCA and FraX genetic diagnostics. For the six samples with both ONT and Illumina data available, data were analyzed using MOLGENIS VIP v7.9.1 [28] for the nine SCA genes and *FMR1*. In short, Minimap2 v2.27 [33] was used for alignment against the NCBI GRCh38 reference genome, followed by SNV and indel calling using Deepvariant v1.6.1 [36]. In the WES data, duplicate reads were marked using samtools markdup [37]. In addition, for comparison, VCF files produced in previous diagnostics, that had variants called using the GRCh37 reference genome, were lifted over to GRC38 using UCSC LiftOver (https://www.genome.ucsc.edu/cgi-bin/hgLiftOver (accessed on 19 March 2025)).

For all six samples, we retrieved the variants identified in one of the exons of the 10 genes (Appendix A). This was done by selecting all regions in the NCBI Refseq build 38 MANE select, with exons at each side flanked with 50 bp, overlapping with one of the gene regions. For variants in this region, we compared the ONT LRS calls with those made in the WES data. When there were discrepancies, we determined their cause, using the quality and depth criteria shown in the VCF and BAM files.

### 4.11. Methylation Calling

We used standalone Dorado 0.5.3 (https://github.com/nanoporetech/dorado (accessed on 19 March 2025)) to perform methylation calling, using POD5 files as input data and creating aligned BAM files for base calls with modifications. The Dorado model used was as follows: r10.4.1, e8.2, 400 bps, and hac v4.2.0. Dorado uses minimap2 [33] to align the reads. Modkit 0.2.6 (https://github.com/nanoporetech/modkit (accessed on 19 March 2025)) was used for post-processing base modifications after base calling. It creates summary counts of methylated and unmethylated reads in an extended bedMethyl format. Methplotlib 0.2.1 [38] was then used to visualize the methylation information from the bedMethyl files. Bam files were viewed in IGV-viewer [35], and the methylation status was highlighted by coloring alignments by base modification (5mC) and assessed around the *FMR1* promoter locations chrX:147911902-147911961 and chrX:147912021-147912080 [genecards.org].

## Figures and Tables

**Figure 1 ijms-26-02850-f001:**
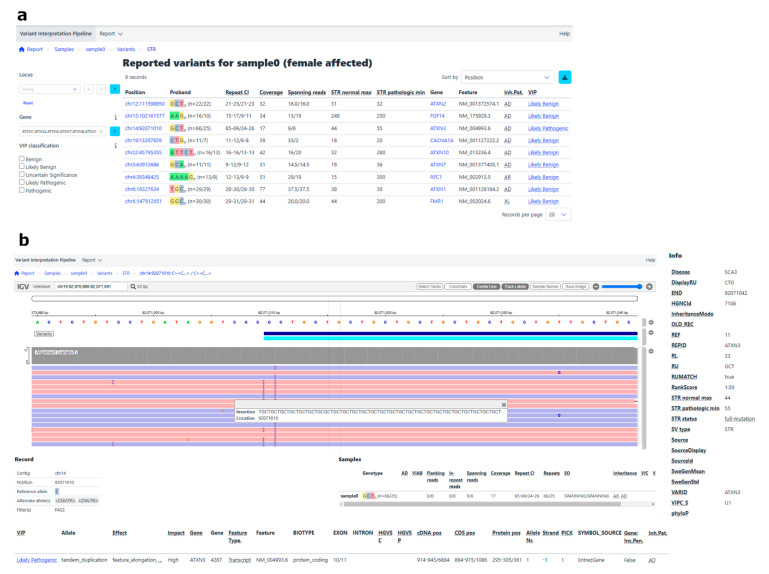
MOLGENIS VIP report for a fictitious sample. The report consists of two sections. (**a**) Upon opening the report, the first section shows an overview of all the STRs in the panel, along with allele repeat lengths, the suggested classification, and the number of reads supporting the call (coverage). The block at the left allows the user to specify that only STRs with a specific classification (e.g., VUS, LP, and P) are shown, which in this example would mean only the *ATXN3* repeat is shown. For each variant, the most important metrics are shown, including the repeat confidence interval (CI), coverage, and reads spanning the entire repeat for both alleles. (**b**) The second section can be opened by clicking on a specific variant. It provides detailed information for this variant. The top section is an IGV view of the repeat region. The bottom section gives all available annotations for this variant. The example shows the variant view for the *ATXN3* repeat.

**Figure 2 ijms-26-02850-f002:**
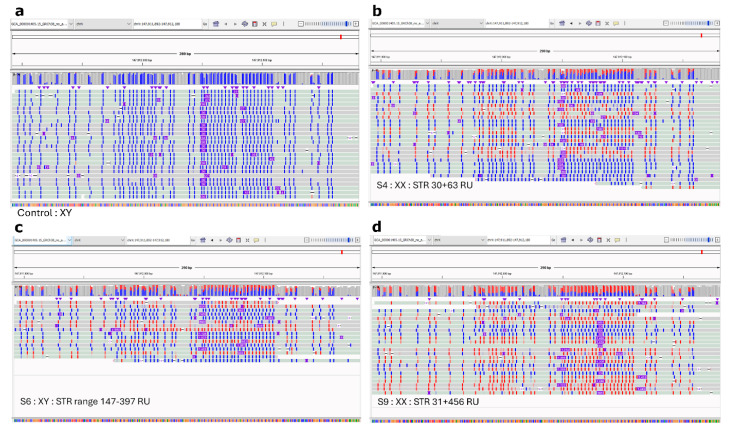
Methylated (red) and unmethylated (blue) *FMR1* promoter regions. For each sample, the region depicted is chrX:147,911,892-147,912,180. (**a**) Control XY, no repeat expansion; (**b**) control XX, no repeat expansion, (**c**) affected male, XY, mosaic repeat expansion, and (**d**) female carrier, XX, repeat expansion.

**Table 1 ijms-26-02850-t001:** The number of reads supporting STR calls for all genes in each of the validation samples (S1–S12).

Sample	*ATXN2*	*ATXN8*	*FGF14*	*ATXN3*	*CACNA1A*	*ATXN10*	*ATXN7*	*RFC1*	*ATXN1*	*FMR1*	Average (Autosomal Genes)
S1 (m)	28	38	29	31	26	41	37	18	22	11	30
S1_1	16	29	18	14	13	24	17	11	13	6	17
S1_2	12	9	11	17	13	17	20	7	9	5	13
S1 (r)	21	12	15	22	17	16	13	8	18	15	16
S2 (m)	38	30	30	40	14	37	32	33	27	16	32
S2_1	14	14	15	21	5	17	13	15	11	9	14
S2_2	24	16	15	19	9	20	19	18	16	7	18
S2 (r)	18	22	17	13	15	24	26	16	32	11	21
S3 (m)	15	22	20	16	18	28	17	22	30	22	20
S3_1	10	15	14	20	15	13	16	18	22	16	16
S3_2	15	22	20	16	18	28	17	22	30	22	20
S4	30	29	32	27	16	33	34	13	36	22	27
S5	26	32	27	27	25	29	26	36	33	6	29
S5 (r)	26	32	27	27	25	29	26	36	33	6	29
S6	25	23	27	29	39	27	33	31	31	11	30
S7	23	14	8	15	16	19	16	20	19	10	16
S8	22	23	19	17	18	18	21	17	17	11	19
S9	22	35	19	31	35	24	27	20	18	20	25
S10	29	35	35	30	23	38	23	32	30	13	30
S11 (m)	83	79	101	89	74	91	99	88	75	59	86
S11_1	40	40	54	46	30	43	44	42	46	34	42
S11_2	43	39	47	43	44	48	55	46	29	25	44
S12	25	30	36	34	23	29	34	30	23	25	29

m: Merged (all data of two combined flow cells was processed). _1: data of the first flow cell. _2: data of the second flow cell. r: Replicate (a separately processed duplicate sample).

**Table 2 ijms-26-02850-t002:** STR lengths of the control variants as determined with LRS.

Sample	*FGF14*	*ATXN1*	*ATXN2*	*ATXN3*	*CACNA1A*	*ATXN7*	*FMR1*	Note
S1	304/348 ^				12/12			
S2		31/54						
S3		30/42 ^^#^	21/22 ^	15/29 ^	7/11	10/10		
S4							30/63	
S5		28/30 ^+^^	22/22	23/23	12/12	11/11		
S6							223 ^†^	*FMR1* range 147–397
S7				28/69 *				
S8								
S9							31/456	
S10								
S11		29/29	22/22	23/28^	11/11	10/12 ^		
S12							31/393	

STR lengths are shown as allele1/allele2 in a number of repeat units (RU). No symbol: LRS result within 2 RU difference with a diagnostically reported STR length. * 3 RU difference in STR length in comparison to diagnostic result. ^+^ 4 RU difference in STR length in comparison to diagnostic result. ^†^ Median length 223 RU (range 147 to 397); diagnostically reported range 213 to >300. ^ In the case of discrepancies between replicates, the median (rounded up) number of RU is shown. ^#^ No CAT interruption seen in the IGV viewer. *FGF14* S1 (merged): 303/348, S1_1: 300/350, S1_2:304/342, S1_replicate: 305/348; *ATXN1* S3 (merged): 30/41, S3_1 and S3_2 30/42, S5: 26/30, S5_replicate 30/30; *ATXN2* S3 (merged): 21/21, S3_1 22/22, S3_2 20/22; *ATXN3* S3 (merged) and S3_2: 15/27, S3_1 15/26, S11 (merged) 20/24, S11_1 20/24, S11_2 20/25. *ATXN7*: S11 (merged) 11/11, S11_1 9/12, S11_2 10/12.

**Table 3 ijms-26-02850-t003:** The classification of STR lengths of patient samples with a VUS or LP of at least one allele.

	Reported Variant 1				Reported Variant 2			
Sample	Gene	STR Length (Allele1/Allele2)	Classification	Repeat Motif or Presence of Interruptions?	Gene	STR Length (Allele1/Allele2)	Classification	Repeat Motif
S1 (m)	*FGF14*	303/348	LP/LB *	GAA/GAAGGA	*RFC1*	134/134	VUS/VUS	AAAGG/AAGGG
S1_1	*FGF14*	300/350	LP/LB *	GAA/GAAGGA	*RFC1*	136/136	VUS/VUS	AAAGG/AAGGG
S1_2	*FGF14*	304/342	LP/LB *	GAA/GAAGGA	*RFC1*	129/129	VUS/VUS	AAAGG/AAGGG
S1 (r)	*FGF14*	305/348	LP/LB *	GAA/GAAGGA	*RFC1*	134/134	VUS/VUS	AAAGG/AAGGG
S2 (m)	*ATXN1*	31/54	LB/LP		*RFC1*	105/374	LB †/LP	AAAG, AAAAG/AAGGG
S2_1	*ATXN1*	31/54	LB/LP		*RFC1*	104/362	LB †/LP	AAAG, AAAAG/AAGGG
S2_2	*ATXN1*	31/54	LB/LP		*RFC1*	107/374	LB †/LP	AAAG, AAAAG/AAGGG
S2 (r)	*ATXN1*	31/54	LB/LP		*RFC1*	106/378	LB †/LP	AAAG, AAAAG/AAGGG
S3 (m)	*ATXN1*	30/41	LB/LP		*RFC1*	11/528	LB/LP	Allele 2 AAGGG
S3_1	*ATXN1*	30/42	LB/LP		*RFC1*	11/524	LB/LP	Allele 2 AAGGG
S3_2	*ATXN1*	30/41	LB/LP		*RFC1*	11/528	LB/LP	Allele 2 AAGGG
S4	*FMR1*	30/63	LB/VUS		*RFC1*	142/142	VUS	AAAAG with AAAG interruptions
S5								
S5 (r)								
S6	*FMR1*	223	LP	GGT interruptions	*RFC1*	15/141	LB/LB †	AAAAG with AAAG interruptions
S7	*ATXN3*	25/66	LB/LP					
S8					*RFC1*	11/122	LB/LB †	Allele 2 AAAAG
S9	*FMR1*	31/456	LB/LP	GC, GCA and GAC interruptions	*RFC1*	12/114	LB/VUS	AAAAG
S10								
S11 (m)								
S11_1								
S11_2								
S12	*FMR1*	31/393	LB/LP		*RFC1*	12/112	LB/VUS	Allele 2 AAAAG

Candidate VUS and LP repeats as detected by the MOLGENIS VIP, with reclassification based upon a visual inspection of RU. Repeats labelled LB are shown if it initially was classified as VUS or LP based on repeat length or if the other allele is (or was) classified as VUS or LP. m = data of two flow cells are merged. r = replicate sample. * Manual classification based on repeat motif. Automatic classification was LP. † Manual classification based on repeat motif. Automatic classification was VUS.

**Table 4 ijms-26-02850-t004:** The number of variants with a coverage >3 with both LRS and NGS and detected on either one of the platforms.

Sample	Number of Indels			Number of SNVs		
	Concordant Both Platforms	LRS Only	WES Only	Concordant Both Platforms	LRS Only	WES Only
S1	4	1	0	21	12 *	1
S3	4	0	0	41	1	1
S4	5	0	0	28	0	0
S5	5	0	0	25	0	0
S8	3	0	0	33	0	2
S11	4	0	0	32	0	0

Discrepant calls are only reported when they are not directly next to or inside a homopolymer or other repeat and not part of another called variant. * Not called in the NGS data because tagmentation resulted in reads starting at the same position, leading samtools markdup to label them as duplicate reads.

## Data Availability

The datasets for this article are not publicly available because of concerns regarding patients’ anonymity. Our data are considered as a personal identifier. Requests to access the datasets from qualified researchers should be directed to the corresponding author. There are restrictions on a qualified researcher accessing the data (non-commercial use only and requiring a Data Usage Agreement).

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
