# Peer review of "Nanopore Long-Read Sequencing as a First-Tier Diagnostic Test to Detect Repeat Expansions in Neurological Disorders"

_ijms, 2025, doi:10.3390/ijms26072850_

Round 1

Reviewer 1 Report

Comments and Suggestions for Authors

In this work by et al, authors explore a novel approach (one-test-fits-all) for implementating of Oxford Nanopore Technologies long-read sequencing in the diagnostic workflow for detecting short tandem repeat (STR) expansions in neurological disorders, specifically SCA and FraX. Therefore, the article addresses a topic of great relevance in the field of medical genetics. The results show high concordance with existing diagnostic methods.

Some comments:

- Larger copy number variation and structural variants other than STRs were not tested in this study. The authors justify that this is not need to be included in this test by citing a previous study in Dutch patients, which found them to be a very rare cause of SCA. However, taking into account the potential genetic variability across populations, this could limit the generalizability of the method (especially if the authors claim that “this method can be generalized to other diseases”)

- Were comparisons or validations performed against Southern blot, which is often considered the gold standard for detecting large repeat expansions?

- On the other hand, the authors acknowledge certain limitations for this study, such as the small sample size and that a higher coverage is needed to improve the accuracy. This raises questions about the scalability and cost-effectiveness of the method in clinical settings, where high coverage may not always be practical.

- Given the specific characteristics of this technology, it would be interesting to include considerations about cost, time and necessary resources for clinical implementation (including the need for manual reanalysis of some data). These aspects are critical to determining whether this method can be widely adopted in diagnostic laboratories

- The study is based on a small cohort of only 12 cases. While the results are promising, a larger and more diverse cohort would be necessary to confirm the reproducibility and robustness of the findings across different laboratories and populations.

Author Response

Comment 1: Larger copy number variation and structural variants other than STRs were not tested in this study. The authors justify that this is not need to be included in this test by citing a previous study in Dutch patients, which found them to be a very rare cause of SCA. However, taking into account the potential genetic variability across populations, this could limit the generalizability of the method (especially if the authors claim that “this method can be generalized to other diseases”)

Response 1: Thank you for this comment. We changed “Finally, we previously demonstrated that copy number variations (>50 kb) are only a very rare cause of SCA [28] and do not need to be included in this test.” in “Finally, we previously demonstrated that in the Netherlands, copy number variations (>50 kb) are only a very rare cause of SCA [28]. Therefore, we did not include detection of these types of variants in this test. However, these types of variant will be important for other diseases and the detection and annotation of these variants can easily be added to our modular pipeline [32].”  (line 292-297 of the revised manuscript)”

Comment 2:  Were comparisons or validations performed against Southern blot, which is often considered the gold standard for detecting large repeat expansions?

Response 2: We agree with the reviewer that Southern blot is a good method for STR detection. However, we have used PCR fragment analysis for SCA and the Amplidex PCR CE FMR1 kit for FMR1 repeat expansion detection as our gold standard. Both methods are very well validated and can function as gold standard. Therefore, we believe there is no need for additional testing with Southern blot.

Comment 3 and 4:

- On the other hand, the authors acknowledge certain limitations for this study, such as the small sample size and that a higher coverage is needed to improve the accuracy. This raises questions about the scalability and cost-effectiveness of the method in clinical settings, where high coverage may not always be practical.

- Given the specific characteristics of this technology, it would be interesting to include considerations about cost, time and necessary resources for clinical implementation (including the need for manual reanalysis of some data). These aspects are critical to determining whether this method can be widely adopted in diagnostic laboratories

Response 3: We added “Depending on the already available resources of laboratories, this will be or in the near future become cost-effective because more variant types will be detected with one single technique.” to the manuscript (line 202-204) to discuss this in more detail.

Regarding the need for manual reanalysis, we are working on automatic detection of motifs and added “However, future developments will allow for automatic motif detection abolishing the need for manual reanalysis thereby allowing wider adoption of the technique.” to the manuscript (line 286-288).

Comment 4: The study is based on a small cohort of only 12 cases. While the results are promising, a larger and more diverse cohort would be necessary to confirm the reproducibility and robustness of the findings across different laboratories and populations.

Response 4: Thank you for this suggestion, we added “A larger and more diverse cohort would be necessary to confirm the reproducibility and robustness of the findings across different laboratories and populations” to the manuscript (line 204-206).

Reviewer 2 Report

Comments and Suggestions for Authors

This work represents a significant contribution to the field of genetic diagnostics, particularly for neurological disorders caused by expansions of short tandem repeats (STRs). However, some comments follow:

- The introduction is well organized, but could be improved with a brief description of the limitations of current techniques, such as SRS and PCR. 

-In the results, the authors present a mismatch in the detection of STR length in Sample 5. It would be useful to discuss this result, indicating how the bias found could be solved. 

- The discussion is robust, but could be enriched with a more in-depth comparison with other similar studies using the LRS for STR, SNV, Indel and methylation analysis.

Author Response

Comment 1: - The introduction is well organized, but could be improved with a brief description of the limitations of current techniques, such as SRS and PCR.

Response 1: We agree with the reviewer this is an important topic. We elaborated further on this topic in line 37-46 the manuscript. We changed “However, the sizes reported are inaccurate [6,7], so several PCR techniques are currently used to accurately measure the expansion length and motif” in “However, the sizes reported are inaccurate [6,7], so several different PCR techniques are currently used to accurately measure the expansion length and motif requiring always multiple PCR reactions per patient.” to describe the limitations of the current techniques more clear.

Comment 2: In the results, the authors present a mismatch in the detection of STR length in Sample 5. It would be useful to discuss this result, indicating how the bias found could be solved.

Response 2:

We agree with the reviewer that adding such information is useful and therefore added the following to the manuscript: “This was further supported by the fact that there were 2 and 33 reads supporting the call of respectively the 26 and 33 repeat units. This does not sufficiently support a heterozygous call with 26 repeat units when the coverage is 35x. In the analysis pipeline these outlier calls can be avoided by increasing the thresholds for read support depending on the coverage” (line 92-95).

Comment 3: The discussion is robust, but could be enriched with a more in-depth comparison with other similar studies using the LRS for STR, SNV, Indel and methylation analysis.

Response 3: We thank the reviewer for this suggestion. We changed “Our results and conclusions are in line with earlier research performed on detection of STRs using ONT LRS adaptive sampling [9,10].” in “Our results and conclusions are in line with earlier research performed on detection of STRs, SNVs, Indels and methylation using ONT LRS adaptive sampling in highly similar patient groups [9,10]. Compared to these previous studies, we used automated STR-calling and classification which brings LRS-based STR-analysis closer to clinical use.” (line 207-210).